## Research Article

coastal resilience; sea-level rise; coastal flooding; flood management; washover

**Corresponding author:**
Hayden Tackley;
Email: hayden.tackley@dal.ca

# First-order assessment of flood vulnerability in the Chignecto Isthmus, Atlantic Canada

Hayden A. Tackley ⓘ, Bay Berry, Nicole K. LeRoux and Barret L. Kurylyk

Department of Civil and Resource Engineering & Centre for Water Resources Studies, Dalhousie University, Halifax, Canada

## Abstract

The Chignecto Isthmus is the sole land connection between Nova Scotia and mainland Canada, supporting national trade, agriculture and transportation. Much of this low-lying corridor is protected by aging dikes that are increasingly vulnerable to compound flooding from tides, storm surges and sea-level rise. This study combines static flood modeling and GIS-based land use classification to evaluate the elevation-based flood exposure of infrastructure and agricultural land. A planar water surface modeling approach validated with differential GPS measurements was applied to a 1-m-resolution digital elevation model. Results indicate that water levels in the adjacent basin can reach within 1 m of the mean dike crest elevation during spring tides. Planar surface modeling scenarios demonstrate that relatively modest increases in water level beyond this threshold could result in inundation, affecting thousands of hectares of cropland and hundreds of hectares of developed land, along with critical transportation infrastructure. This exposure has the potential to disrupt agricultural productivity, rural livelihoods, groundwater quality and interprovincial supply chains across the isthmus. While simplified, this analysis highlights the diminishing safety margin afforded by existing dikes, underscores the need for more detailed scenario-based modeling and reinforces the importance of proactive adaptation planning to safeguard this nationally significant corridor.

## Impact statement

This study highlights the Chignecto Isthmus as a nationally critical corridor in Eastern Canada. The region's low elevation, deteriorating flood protection infrastructure and concentrated elevation-based flood exposure to high coastal water levels make it especially vulnerable to sea-level rise and extreme storm events. Beyond regional significance, our findings have broader implications for low-elevation coastal zones worldwide, where adaptation must account for overlapping land uses, ecological trade-offs and multi-jurisdictional governance. The patterns of exposure to high water levels identified in this study draw attention to questions of responsibility and cost when responding to sea-level rise, framing them as key issues for future adaptation planning and research.

## Introduction

Coastal flooding is a significant risk associated with oceanic climate change due to rising sea levels, more intense storms and the reduction of natural protective buffers (Gilman et al., 2008; Oppenheimer et al., 2019). Low-lying coastal areas and estuaries across the globe are increasingly susceptible to inundation (Kulp and Strauss, 2019; Edmonds et al., 2020; Tackley et al., 2025a). Recent studies estimate that by the end of the century, this issue may affect hundreds of millions of people, put infrastructure valued in trillions of dollars at risk (Hinkel et al., 2014) and, in some cases, force the migration of low-elevation populations (McGranahan et al., 2007). This growing threat underscores the need to better understand and adapt to changes to vulnerable coastal areas globally.

One such vulnerable area is the Chignecto Isthmus, a narrow stretch of land connecting the province of Nova Scotia to mainland Canada, that serves as a vital corridor for transportation, trade and food production. With only one major overland route linking the province to the rest of the country, the isthmus supports approximately 35 billion dollars (CAD) in annual freight movement and provides access to the Port of Halifax, a major node in Canada's supply chain (Wood Environment & Infrastructure Solutions, 2022). It also supports critical infrastructure, including power transmission and regional rail lines, all concentrated within a low-elevation floodplain with limited redundancy (Office of the Auditor General of Nova Scotia, 2016; Government of Canada, 2025). Due to its low elevation, the isthmus and the surrounding settlements, infrastructure and agricultural land are vulnerable to flooding, a widely publicized concern in the context of recent severe storms (Tutton, 2019; Patil, 2023; Reynolds, 2023).

The isthmus is shaped by its position between two dynamic coastal systems: the Northumberland Strait to the north and the Bay of Fundy to the south (Figure 1). The Bay of Fundy is home

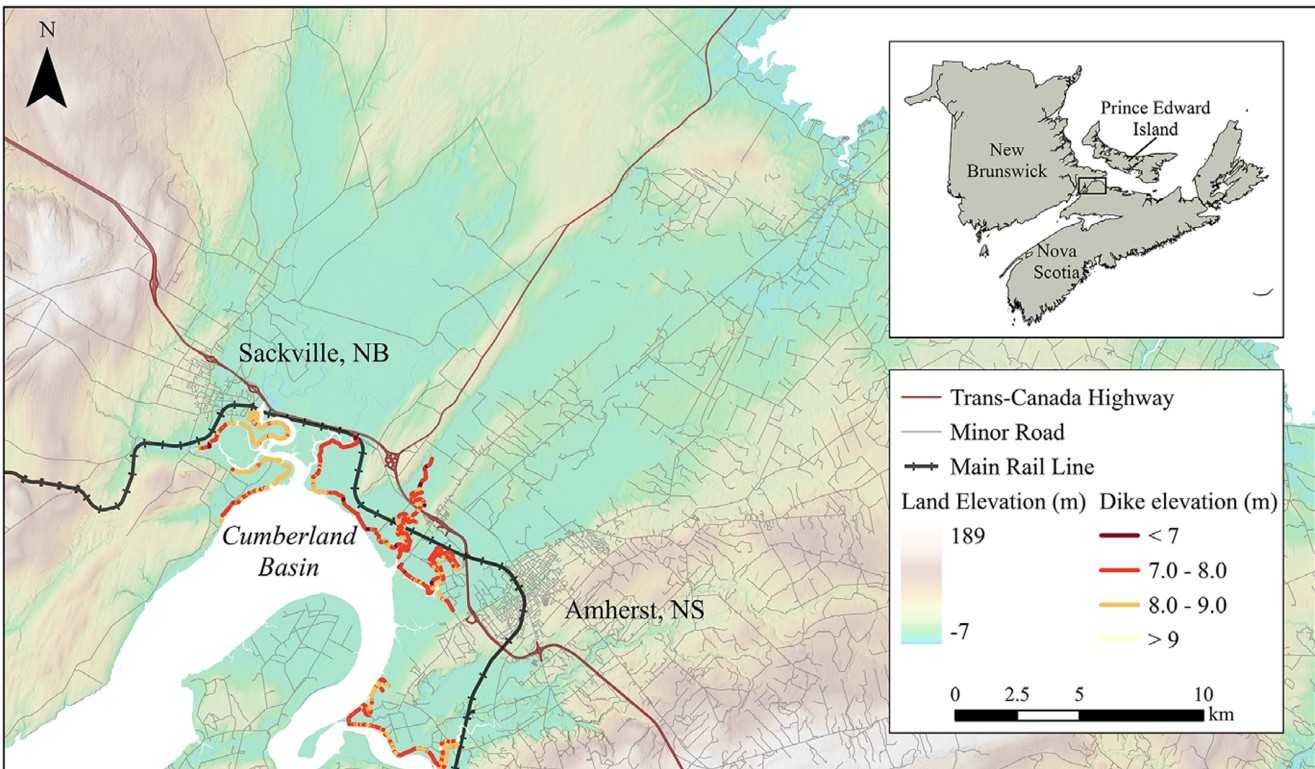

**Figure 1.** Elevation map of the Chignecto Isthmus relative to the Canadian Geodetic Vertical Datum 2013, highlighting key towns and infrastructure corridors. Labeled features include the Trans-Canada Highway (red), main rail line (black) and dike infrastructure (yellow-red). Elevation shading (GeoNB, 2017; Service Nova Scotia and Internal Services, 2019) shows low-lying terrain adjacent to the Cumberland Basin, an inlet to the Bay of Fundy. Inset map shows the regional location within Atlantic Canada.

to the world's highest tidal ranges, with spring tides routinely exceeding 16 m (Desplanque and Mossman, 2004; Shaw et al., 2010). These hypertidal conditions, low-lying terrain and limited natural buffers make the region especially vulnerable to coastal flooding (Archer, 2013). Bathymetric resonance in the bay further amplifies water levels during storm events, and wave–tide–surge interactions are known to increase total water levels in this system (Garrett, 1972). While much of the surrounding landscape is protected by a 35-km network of earthen dikes (Wood Environment & Infrastructure Solutions, 2022), few meet modern design standards, and some sit less than a meter above recent storm-driven water levels (van Proosdij and Page, 2015). Dike conditions vary widely in elevation and structural integrity, with some segments having experienced subsidence, wave overtopping or internal erosion (Webster et al., 2012). With rising seas, more intense precipitation and increasing storm activity, the protective capacity of this infrastructure is being tested with growing frequency (Sherren et al., 2021). Adjacent salt marshes that once provided natural buffering have also been reduced or disconnected from tidal flow, further increasing vulnerability (Virgin et al., 2020).

The isthmus also supports thousands of hectares of farmland, critical bird and wildlife habitat and low-density rural communities with limited emergency access (Webster et al., 2012; Nussey, 2016; Government of Canada, 2022). Groundwater and surface water sources are vulnerable to saltwater intrusion from brackish floodwaters and sea-level rise, and recent studies in the region have shown that saline floodwater and sediment deposits from overtopping can drive long-term degradation of soil and shallow aquifers (Tackley et al., 2023, 2025b). A major flooding event could have cascading consequences for mobility, food systems and regional

economic security (Sherren et al., 2021). Given the large tidal range of the Bay of Fundy, the additional contributions of storm surge and wave setup, and projected relative sea-level rise exceeding 1.6 m by 2100 under a high-emissions scenario (RCP 8.5; James et al., 2015), extreme total water levels in the upper Bay of Fundy may exceed present-day high-tide elevations by more than one meter during rare events, regardless of tidal phase. These compound flood-generating processes motivate the use of a range of modeled water levels to assess how flood exposure is influenced by increasingly severe, yet physically plausible, conditions.

To date, relatively few studies have treated the Chignecto Isthmus as an integrated system of overlapping flood exposure, infrastructure vulnerability and adaptation needs. Although restoration, hydrodynamic modeling and realignment projects have been studied in other parts of the upper Bay of Fundy (e.g., Virgin et al., 2020; Sherren et al., 2021; Burns et al., 2025), the isthmus remains underrepresented in peer-reviewed literature. Much of the existing analysis is found in gray literature, including consulting, government and non-profit reports (e.g., Webster et al., 2012; Rapaport et al., 2017; Wood Environment & Infrastructure Solutions, 2022). While these studies offer valuable insights, there remain opportunities to consolidate and extend their findings through open-access, peer-reviewed research that can serve as a foundation for ongoing research on this site, including more process-based, site-specific flood modeling. This paper combines modeled flood scenarios, land use analysis and field-based elevation data to assess exposure and examine adaptation pathways that could shape the future of this critical corridor. Flood scenarios for this first-order risk assessment are based on a planar surface or 'bathtub' modeling approach (Hooijer and Vernimmen, 2021; Ohenhen et al., 2024), which

assumes static water levels and does not account for dynamic processes such as wave action, storm surge and tidal timing, sediment transport or structural failure of the dike system. Given biases that can be introduced by static flood models (Sanders et al., 2024), we are not attempting to precisely delineate flood zones for a given storm event but rather to gain a high-level understanding of flood risks to guide future, detailed hydrodynamic modeling efforts. Specifically, this study aims to evaluate the extent to which critical infrastructure and agricultural land are exposed to plausible flood scenarios. These exposure results are then used to inform discussion of key barriers and opportunities for implementing adaptation strategies under current governance conditions.

## Study site description

### Background and previous work on the Chignecto Isthmus

Research on the Chignecto Isthmus has expanded in recent years, highlighting both the region's physical vulnerabilities and the socio-political complexities surrounding flood adaptation. Several studies have focused on managed realignment and salt marsh restoration projects in the upper Bay of Fundy, showing promising recovery of vegetation, hydrologic function and sediment accretion following dike breaching (Virgin et al., 2020; van Proosdij et al., 2023; Burns et al., 2025). Restoration has also been linked to enhanced carbon accumulation, offering climate mitigation co-benefits alongside flood protection (Wollenberg et al., 2018). Although many studies address restoration across the Bay of Fundy, a smaller number focus directly on the Chignecto Isthmus as a primary field site or modeling domain. These include long-term monitoring of realigned marshes (van Proosdij et al., 2023), hydrodynamic scenario testing (Burns et al., 2025) and carbon accumulation studies (Wollenberg et al., 2018; Needham et al., 2020).

Recent modeling studies have improved our understanding of tidal dynamics, compound flooding and saltwater intrusion in the Bay of Fundy. Burns et al. (2025) applied Delft3D-FM to simulate post-breach hydrodynamics on a restored floodplain in the Bay of Fundy, providing insight into current speeds, flood extents and restoration feasibility. McLaughlin et al. (2021) and Karsten et al. (2008) demonstrate that tidal resonance in the Bay of Fundy amplifies energy, suggesting that sea-level rise may nonlinearly increase flood risk. These findings validate the use of compound flood scenarios that holistically combine tide, surge and wave setup. Vertical saltwater intrusion is an additional, underrecognized mechanism of degradation in transitional coastal environments (Cantelon et al., 2022). Flood-derived sediments have also been shown to contribute to long-term soil and groundwater salinization along the Bay of Fundy, particularly under managed realignment or storm-driven flooding (Tackley et al., 2025b), reinforcing the idea that subsurface contamination can persist for years or decades, even if floodwaters recede quickly.

Despite these advancements, the Chignecto Isthmus remains underrepresented in peer-reviewed adaptation literature. Sherren et al. (2021) noted that conflicting federal and provincial responsibilities over dike maintenance and flood policy have slowed progress toward coordinated adaptation planning. Uncertainties around long-term funding, land ownership and public support for realignment or retreat compound these governance challenges. Regulatory designations such as the Chignecto Isthmus Wilderness Area Designation (2008) further complicate land use planning, as they restrict specific infrastructure development or flood-mitigation activities in protected zones.

### Tidal dynamics and landscape vulnerability

The Chignecto Isthmus lies at the head of the Bay of Fundy, where tidal amplitudes can exceed 12 m during spring tides (Desplanque and Mossman, 2004). This hypertidal environment, shaped by the bay's narrowing geometry, resonant tidal oscillations and shallow bathymetry, creates expansive intertidal zones and highly dynamic water levels (Supplementary Figure S2). These conditions increase baseline flood exposure and amplify the risk posed by storm surges and waves, particularly when such events coincide with high tide (Singh et al., 2007). Historical events such as the Saxby Gale of 1869 and the Groundhog Day Storm of 1976 illustrate this vulnerability, with storm surges causing extensive flooding and damage across the isthmus (Desplanque and Mossman, 1999).

Much of the isthmus lies at low elevation (Figure 1), and the surrounding terrain has been heavily modified through centuries of land reclamation (Sherren et al., 2021). Most of the coastal transition zones (e.g., salt marshes) in the region were diked beginning in the late 1600s and remain diked today, forming a legacy system of agricultural and infrastructural land use (Shaw et al., 2010). A network of earthen dikes, some originally constructed by Acadian settlers, now protects areas used for farming, transportation and rural settlement. Although some dike segments have been repaired or reinforced over time, many remain below modern flood protection standards and show signs of degradation due to age, subsidence and erosion (Webster et al., 2012).

Geographic Information System (GIS)-based analysis of the region adjacent to the Cumberland basin indicates a mean dike elevation of 7.9 m, with a standard deviation of 0.45 m (Figure 2). This variability reflects the historical development and uneven maintenance of the system, highlighting localized segments that may already sit below critical water levels. Field photographs provided in the Supplementary Material, taken across the isthmus, illustrate

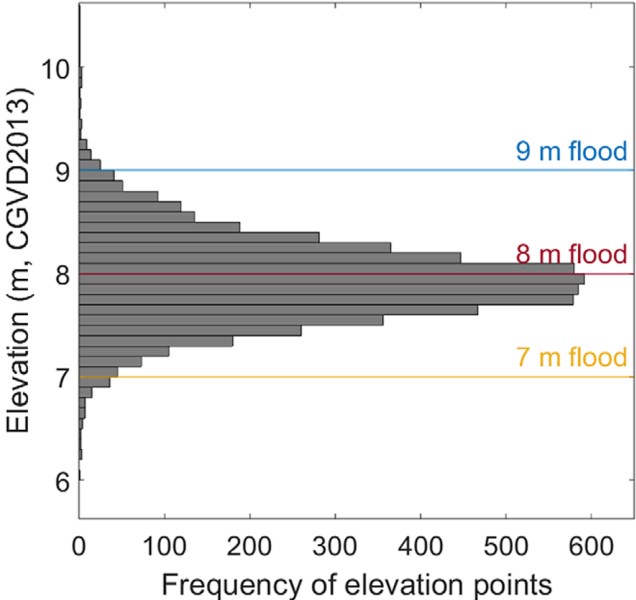

**Figure 2.** Histogram of dike crest elevations across the Chignecto Isthmus, based on points sampled at 10-m intervals along digitized dike polylines. Elevations were extracted from a 1-m-resolution (horizontal) digital elevation model (DEM), referenced to CGVD 2013. The distribution reveals a central tendency near 7.9 m, with notable variability and a long tail toward lower elevations, indicating potential overtopping risk in some segments. Field validation using 54 differential GPS points collected in August 2025 confirmed DEM-derived elevations within a mean error of 0.13 m and root mean square error of 0.24 m.

this elevation variability and the diversity of landforms and land uses within the vulnerable zone, including mudflats, salt marshes, transportation infrastructure and agricultural areas.

## Evaluating flood risk through static water level monitoring

### Flood scenario selection

Maximum water elevations at the Chignecto Isthmus are produced by interacting processes that include astronomical tides, storm surge, wind setup, waves and basin-scale bathymetric resonance. These processes can constructively interfere, and their coincidence is expected to increase as climate-driven changes alter tidal resonance and storm frequency and intensity (Greenberg et al., 2012; Wu et al., 2017). The timing and magnitude of any given event are difficult to quantify because storm impacts depend on seasonal conditions, storm track geometry, wind direction and tidal phase (Quinn et al., 2014). Longer term oscillations, including the 18.61-year nodal tidal cycle, can further modulate baseline water levels and can amplify or attenuate extreme events over decadal timescales (Peng et al., 2019).

To assess the potential severity and spatial extent of flooding across the Chignecto Isthmus, we conducted simple static water level modeling at 7, 8, and 9 m above mean sea level (AMSL). The selected water levels do not represent specific return periods but instead bracket a plausible range of compound high-water conditions for the region within this century to assess the inundation potential. Long-term, continuous water-level data to conduct extreme value analysis (return periods) are not available near the isthmus. However, spring tides in the Cumberland Basin can reach 6.9 m above the Canadian Geodetic Vertical Datum of 2013 (CGVD 2013) under present conditions (Crowell et al., 2025). When combined with storm surge, waves and anticipated sea-level rise, total water levels can plausibly exceed 7 m and approach or surpass 8 m within the coming decades. The 7-m threshold, therefore, represents a present-day compound event that combines spring high tide with a minor storm surge. The 8-m threshold reflects a more severe event or a mid-century scenario in which projected sea-level rise contributes to elevated water levels. The 9-m threshold represents an upper bound, late-century scenario intended to bracket the high end of plausible combined tide, surge and sea-level rise contributions. Given the challenges of defining return periods for the compound tide–surge–wave events in the Bay of Fundy, these thresholds merely provide a realistic range of progressively severe water levels to assess exposure potential. While this method does not simulate dynamic and interacting flow, surge or wave dynamics, it provides a first-order approximation of inundation under plausible high-water conditions, and the three high-water levels considered are intended as screening thresholds rather than probabilistic projections.

Although hydrodynamic models can explicitly simulate many of these interacting processes, their implementation requires detailed information on storm characteristics such as track, timing, surge evolution and wave conditions (Cyriac et al., 2018), as well as bathymetric change, and the exact location and geometry of overtopping or breaching pathways (Lyddon et al., 2018). These inputs are either unavailable or highly uncertain for this region. Planar inundation models are known to differ from hydrodynamic simulations in important ways. Comparative studies have shown that simple bathtub approaches often overpredict the flood extent and can simulate water depths greater than observed because they do not incorporate momentum, drainage pathways, overtopping dynamics or flow constriction (Didier et al., 2019). Ferguson et al. (2022)

emphasized that such approaches omit critical processes like wave setup, overtopping and flood routing, which can strongly influence local impacts. At the same time, planar models avoid the opposite bias in which hydrodynamic models underpredict flooding when breaching or overtopping occurs at locations not explicitly represented in the model boundary conditions (Bates, 2023). In hypertidal environments such as the upper Bay of Fundy, where tide–surge–wave interactions can vary substantially among events, planar models provide a transparent and reproducible means of exploring sensitivity to water levels without relying on uncertain storm-specific parameters. For these reasons, static models are commonly used as screening-level tools for elevation-based flood exposure assessment, with hydrodynamic modeling preferred for detailed scenario analysis in subsequent phases of assessment (Teng et al., 2017).

Because our objective was to evaluate broad patterns of exposure rather than to reproduce specific events, the planar approach is appropriate for identifying areas where more detailed hydrodynamic modeling should be prioritized. Our intent is therefore not to predict exact flood extents, but to highlight areas that are potentially at risk based solely on elevation thresholds. As such, the planar surfaces presented here should be interpreted as screening-level exposure envelopes that may inform future, more detailed hydrodynamic modeling of compound flood behavior in the Chignecto Isthmus.

### Validation and workflow

The three considered flood surfaces were generated using a planar surface modeling approach, intersecting uniform water levels with a 1-m-resolution (horizontal) digital elevation model (DEM) derived from Service Nova Scotia and Internal Services (2019) and GeoNB (2017), referenced to CGVD 2013. Flood surfaces were generated in Esri ArcGIS Pro using a threshold-based raster classification approach. The 1-m-resoultion (horizontal) DEM was reclassified to identify all cells with elevations at or below the 7-, 8- and 9-m thresholds. The resulting binary rasters were then converted to vector polygon features, and areas directly connected to the Bay of Fundy were retained. The resulting flood extent polygons were intersected with land cover and infrastructure layers. No hydrologic subsurface connectivity was imposed, and the method does not simulate flow routing or drainage pathways. The outputs therefore represent elevation-based exposure envelopes that delineate areas that lie below each specified water level, regardless of whether water would reach these locations under a particular storm scenario.

Dikes were treated as part of the existing DEM rather than as additional hydraulic barriers. Crest elevations derived from the DEM reflect the current physical configuration of the dike system. Because breach location, timing and geometry cannot be predicted without detailed hydrodynamic modeling and structural failure assumptions, no artificial breaching scenarios were considered. This approach avoids imposing unverified assumptions about overtopping locations and allows the planar surfaces to represent a generalized assessment of areas that are topographically susceptible to inundation under each water level. The planar method provided a screening-level estimate of exposure based solely on elevation, consistent with standard static inundation approaches used in first-order coastal vulnerability assessments (Teng et al., 2017).

To validate DEM accuracy, 54 spatially distributed differential Global Positioning System (dGPS) points were collected with an Emlid Reach RS2 in August 2025 across the isthmus, including dike crests, roads, rail lines and watercourse crossings. Comparison with

DEM-derived elevations yielded a mean error (bias) of 0.13 m and a root mean square error of 0.24 m, indicating good agreement within the expected vertical accuracy of provincial Light Detection and Ranging (LiDAR) data products (±0.15 m; Service Nova Scotia and Internal Services, 2019). To evaluate the influence of this uncertainty, water-level thresholds of 7, 8 and 9 m can be conceptualized as having an effective range of ±0.24 m. To assess the sensitivity of inundation extent and exposure estimates to DEM vertical uncertainty, a deterministic sensitivity analysis was conducted by varying water-level thresholds by ±0.24 m and recalculating flood extents and all associated exposure metrics for each scenario. Land area near these elevation thresholds may therefore shift between inundated and non-inundated classes depending on the direction of the DEM bias. This uncertainty does not affect the broad spatial patterns observed in the flood extent maps but should be considered when interpreting exposure values near the margins of each scenario. These results confirm that the DEM provides a reliable basis for flood exposure analysis and dike height characterization.

Satellite-derived land cover data were sourced from Natural Resources Canada (Latifovic, 2022) at a 30-m spatial resolution and reclassified into cropland, developed land, wetland and other land uses. Exposed land areas were calculated by intersecting modeled flood surfaces with the land cover layer. To evaluate the spatial variation in crest height and identify segments that may be most susceptible to overtopping, dike crest elevations were extracted from the DEM along the full length of the dike system. A polyline was digitized along the crest of each dike, and points were generated at 10-m intervals along these lines. Elevations at each point were sampled directly from the DEM, resulting in a dataset of 5,694 crest elevation points to be used as comparison thresholds for modeled water levels. Each crest elevation point was classified into one of four exceedance categories: below 7 m, between 7 and 8 m, between 8 and 9 m, and above 9 m AMSL. These categories correspond to the modeled water levels used in this study. Of the 54 dGPS points, 39 were located on dike crests, averaging 7.6 m, broadly consistent with the mean dike crest elevation derived from the 5,594 DEM points (7.9 m).

### Risk assessment results

#### Surface flood exposure

Elevation data indicate a rapidly diminishing safety margin under increasing flood scenarios. Three high-water levels (7, 8 and 9 m AMSL; Figure 2) are considered in this study, all consistent with combined tide, surge and wave setup heights from historical storms and future projections (Desplanque and Mossman, 1999; Greenberg et al., 2012). At 7 m AMSL, water levels remain below the mean dike crest elevation of 7.9 m, though overtopping may still occur at degraded or low-lying segments (Figure 2). At 8 m, overtopping is likely across multiple dike segments, and at 9 m, most of the dike system is exceeded, particularly in the Aulac, Amherst and Sackville corridors. The spatial distribution of dike crest elevations relative to these water level thresholds is shown in Figure 3, highlighting segments where reduced freeboard may increase susceptibility to overtopping under elevated water levels, while recognizing that elevation alone does not capture structural condition or failure mechanisms.

Water elevation in the Cumberland Basin can regularly reach elevations of ~6.5–6.9 m above the datum (CGVD 2013) during spring tides (Crowell et al., 2025; Swatridge et al., 2025), within 1–1.4 m of the mean dike crest along the isthmus. The hypertidal range creates an expansive intertidal platform that rises nearly to the base of the dikes (Supplementary Figure S2), meaning that most infrastructure is built well above the elevation reached by routine high tides. Consequently, spring high tides of less than 7 m do not generally pose a flooding risk on their own, and overtopping concerns emerge only when water levels approach dike crest elevations during compound events (e.g., a spring high tide in phase with a storm surge). Nevertheless, establishing a representative mean high tide in the Cumberland Basin presents considerable difficulty due to the pronounced impact of local bathymetric variability, tidal resonance and nonlinear tidal dynamics combined with the general lack of federal or provincial tidal monitoring in the upper Bay of Fundy. These factors can result in significant deviations from standard tidal benchmarks. It is, therefore, reasonable that when tides, surges and waves coincide, water elevations may reach levels presented here, especially when future sea-level rise is considered. These complexities further justify evaluating a range of high-water scenarios rather than a single level.

In addition to direct overtopping, estuarine channels and tidal rivers can act as hydraulic 'short circuits' (Hingst et al., 2023), allowing saline floodwaters to propagate far inland through existing drainage networks or behind aging dike infrastructure. These pathways can introduce saltwater into areas that appear protected based solely on topographic elevation, thereby compounding the risk of subsurface contamination and long-term soil degradation.

To quantify land use exposure, modeled water surfaces were intersected with classified land cover data. Table 1 summarizes the estimated extent of cropland, developed land and linear infrastructure exposed at each water level. Under the 7-m scenario, approximately 8,128 (±111) hectares of cropland and 425 (±29) hectares of developed land fall within the modeled flood zone. By 9 m, exposure increases to approximately 8,771 (±65) hectares of cropland and 620 (±23) hectares of developed land (Figure 3, Table 1). Spatially, most of the increase in inundated area under subsequent flood levels occurs in the northern portion of the study area, while exposure in the southern section, where most towns and infrastructure are concentrated, changes only marginally among scenarios. Consistent with this pattern, sensitivity to DEM uncertainty is greater for the 7-m scenario than for the 8- and 9-m scenarios, reflecting the presence of extensive low-relief terrain near the 7-m elevation threshold in the north, whereas higher relief areas in the south remain above both modeled water levels. Because the DEM uncertainty is on the order of 0.24 m, land areas located close to the modeled thresholds may be sensitive to small changes in elevation. Although this uncertainty may shift the exact area of inundation by several percent, the overall exposure patterns remain consistent. Based on an average operating revenue of approximately $2,480 (CAD) per hectare (Government of Nova Scotia, 2022), roughly $22 million in annual agricultural output may be at risk under these scenarios, not including recovery costs or long-term salinization impacts. These lands also provide critical provisioning and cultural services as well as contributions to local food systems, rural livelihoods and Acadian heritage (Sherren et al., 2021), which would be difficult to restore following repeated saltwater flooding events.

Infrastructure is similarly vulnerable to flooding (Table 1). The Trans-Canada Highway and Canadian National Railway line traverse low-lying terrain near the towns of Aulac, Amherst and Sackville (Figure 3), with some segments lying less than one meter above recent storm-driven water levels. While these routes remain largely unaffected at 7 m AMSL, widespread overtopping is expected under 9-m scenarios, especially where dike protection is limited or absent. Power transmission lines and substations also cross the corridor, compounding the potential for cascading

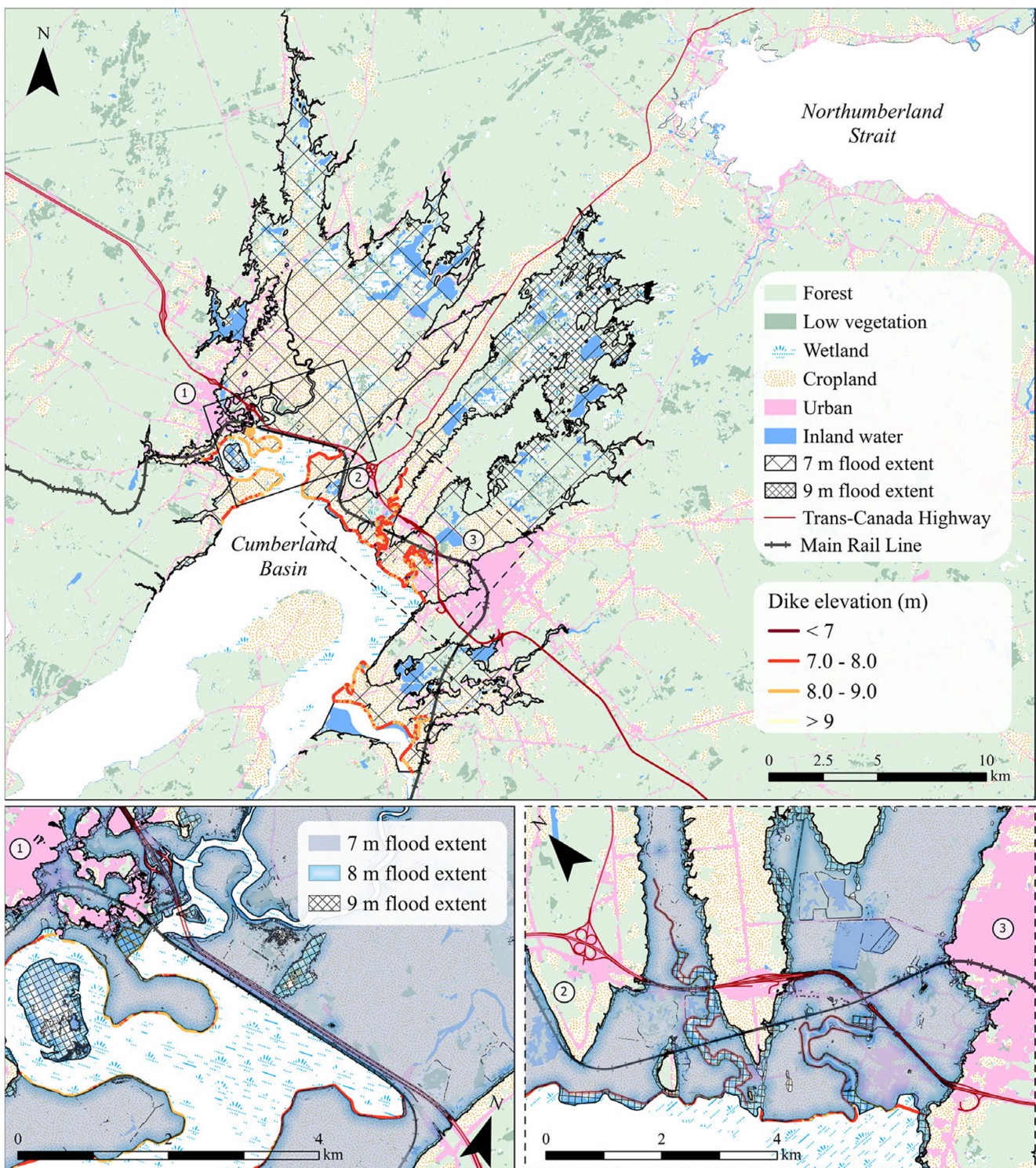

**Figure 3.** Land use classification across the Chignecto Isthmus based on 30-m resolution data from Natural Resources Canada (Latifovic, 2022). Major categories include cropland, developed land, wetland and forest. This map supports the flood exposure analysis by showing the distribution of land uses relative to low-elevation areas identified in Figure 1. Critical infrastructure corridors, such as the rail line and Trans-Canada Highway, intersect both cropland and developed zones at risk of flood exposure under modeled high-water scenarios. Numbers represent the built-up areas of Sackville (1), Aulac (2) and Amherst (3). Inset maps provide corridor-scale views highlighting the spatial relationship between land use, transportation infrastructure and modeled flood extents in areas where exposure is most concentrated.

infrastructure disruptions (Webster et al., 2012). Linear infrastructure exposure was evaluated by intersecting transportation centerlines with the flood polygons derived from the planar surface flood model. Road and rail centerlines were clipped using each flooded area polygon, and the length of all resulting inundated segments was calculated in ArcGIS Pro. This method identifies portions of each corridor that lie below modeled water levels. Because centerlines do not represent the full width or elevation variability of transportation

**Table 1.** Estimated exposure of land use and linear infrastructure under three modeled flood scenarios (7, 8 and 9 m above mean sea level)

| Flood level (m AMSL) (±) | Total area inundated (ha) (±) | | Cropland exposed (ha) (±) | | Cropland exposed (% of flooded area) | Developed land exposed (ha) (±) | | Developed land exposed (% of flooded area) | Road inundation (m) (±) | | Rail inundation (m) (±) | |
|---|---|---|---|---|---|---|---|---|---|---|---|---|
| 7 (0.24) | 17,102 | (1,206) | 8,128 | (111) | 47.5 | 425 | (29) | 2.5 | 488 | (486) | 3,270 | (1055) |
| 8 (0.24) | 19,731 | (353) | 8,483 | (73) | 43.0 | 525 | (23) | 2.7 | 5,218 | (904) | 8,450 | (864) |
| 9 (0.24) | 21,092 | (320) | 8,771 | (65) | 41.6 | 620 | (23) | 2.9 | 6,954 | (249) | 11,502 | (249) |

*Note:* Land cover data were derived from 30-m-resolution classification datasets (Latifovic, 2022), and flood extents were generated using a static planar surface model intersected with a 1-m digital elevation model (GeoNB, 2017; Service Nova Scotia and Internal Services, 2019). Road and rail exposure was calculated using line feature data intersected with modeled flood surfaces in ArcGIS Pro. The variability (±) exposure values were calculated from the RMSE (0.24 m) between measured dGPS points and the provincial DEM files.

embankments and because the planar model does not simulate flow routing, the reported lengths should be interpreted as a first-order estimate of exposure rather than an exact prediction of operational failure.

Under the 7-m scenario, approximately 488 (±486) m of road and 3,270 (±1,055) m of rail lie within the floodplain. The comparatively large uncertainty range at this water level can be attributed to small elevation differences near the lower flood threshold, reflecting the influence of current constructed elevations. These figures rise significantly at higher water levels: by 8 m, an estimated 5,218 (±904) m of road and 8,450 (±864) m of rail are exposed. At 9 m, exposure increases to 6,954 (±249) m of road and 11,502 (±249) m of rail. These findings highlight the sensitivity of linear infrastructure to even modest increases in water level and underscore the importance of targeted protection and redundancy planning. Because rail and highway systems function as continuous corridors, the failure or inundation of even a short segment can render the entire route inoperable until access is restored. Moreover, alternative routing options around the Chignecto Isthmus are minimal, making the region a critical single point of failure in the national transportation network. Although the modeled scenarios will not cause permanent inundation, the isthmus facilitates approximately $100 million (CAD) in daily trade (Government of Nova Scotia, 2024). As such, even brief disruptions could carry significant economic consequences for the region.

Although the flood scenarios presented here are simplified, they offer an elevation-based representation of potential exposure across the region. The static 'bathtub' model assumes uniform water levels and does not account for wave overtopping, flow velocities or dike failure. As such, localized flooding patterns could deviate from these inundation maps, especially in areas with degraded infrastructure or poor drainage. Elevation data were derived from provincial LiDAR surveys, with an estimated vertical uncertainty of ±0.15–0.24 m (Service Nova Scotia and Internal Services, 2019), which introduces ambiguity in threshold-based assessments near dike crests. Similarly, the 30-m-resolution land cover data may obscure small-scale features such as drainage ditches or berms. This interpretation is consistent with large-scale assessments that employ static models for first-order screening but note their inability to resolve local extremes and dynamics such as wave overtopping or channelized flow (Vousdoukas et al., 2016; Hinkel et al., 2021).

This exposure-based framing aligns with other evaluations of planar surface models. Didier et al. (2019) showed that such approaches frequently exaggerate flood extent and depth, while Ferguson et al. (2022) demonstrated that they cannot capture important temporal and hydraulic processes. Nevertheless, both studies confirm the value of simple static methods for regional-scale risk assessments, provided their limitations are acknowledged.

Interpreting the scenarios as present and future conditions provides context for the spatial progression of inundation. The 7-m case approximates a high-water event that could occur under present-day extreme tide and surge combinations, while the 8- and 9-m cases represent increasingly severe conditions that may become more likely as sea levels rise over the coming decades. Future applications on the Chignecto Isthmus could benefit from comparing the elevation-based exposure envelopes presented here to hydrodynamic modeling to better resolve flow pathways, overtopping dynamics, and compound surge–wave and river-basin interactions.

### *Secondary impacts*

Subsurface salinization represents an additional and longer-term pathway of impact that can follow even short-duration coastal flooding. Although the static inundation scenarios presented here describe only surface water extent, the introduction of more saline water into estuaries and dikeland agricultural soils can initiate vertical and lateral salt migration that persists long after surface water has drained. Previous work in the upper Bay of Fundy has shown that salt deposited on the soil surface can infiltrate rapidly into the vadose zone, elevating porewater salinity for multiple growing seasons and reducing crop productivity (Tackley et al., 2023). Capillary rise can further redistribute salts upward under dry conditions, reinforcing salinity stress in root zones (Chen et al., 2022). Even short-duration flood events have been shown to deposit saline sediments that trigger vertical saltwater intrusion into soils and shallow aquifers (Tackley et al., 2023). These processes can persist for years after surface waters have receded, altering water chemistry and reducing agricultural productivity (Elsayed and Oumeraci, 2018; LeRoux et al., 2023). In areas where overtopping or channelized flow conveys brackish water behind dike alignments, shallow unconfined aquifers may also be affected (Tackley et al., 2024). Flood-derived salts can mix with groundwater, travel laterally or occupy preferential flow paths in sandy or reworked agricultural soils (Tackley et al., 2024). Such processes are typically invisible in surface-based mapping yet may influence land use decisions, drainage management and freshwater availability for years or decades following an event.

Subsurface salinization also has implications for buried and partially buried infrastructure. Increased soil salinity accelerates corrosion of metal pipes, culverts and concrete reinforcement, potentially reducing the service life of infrastructure in areas where flooding occurs repeatedly (Habel et al., 2024; Kurylyk et al., 2025). These belowground impacts underscore that surface inundation represents only the initial phase of hazard expression; secondary hydrogeologic and geochemical responses can substantially lengthen recovery timelines.

These subsurface processes are discussed here as qualitative impact pathways informed by prior field studies, rather than as modeled outcomes of the static flood analysis. While quantifying these subsurface dynamics is beyond the scope of this first-order static modeling study, recognizing their potential occurrence highlights the need for integrated monitoring and assessment in flood-prone areas of the isthmus. Future work could incorporate shallow piezometer networks, soil salinity sampling or coupled surface–subsurface numerical simulations to better characterize how repeated overtopping events may alter soil and groundwater conditions over time.

## Adaptation challenges and opportunities

As climate-driven flood risk increases, the Chignecto Isthmus faces urgent decisions about managing its aging flood infrastructure and increasingly vulnerable landscape. Traditional responses – primarily raising or repairing existing dikes – have provided short-term protection, but their effectiveness is declining as storms intensify and sea levels rise. Many of the region's dikes were not designed to withstand the magnitude or frequency of compound flood events now anticipated under changing climate conditions (Lieske and Bornemann, 2011). The spatial flood extent and dike crest elevation patterns presented in Figure 3 highlight that exposure is unevenly distributed across the isthmus, with low-lying diked landscapes and critical transportation corridors experiencing the greatest convergence of flood risk, particularly within the Ahmerst–Aulac–Sackville corridor. Continued reliance on linear defenses may delay failure, but it does not reduce long-term exposure across the broader system. The spatial patterns identified in this study highlight that some segments of the dike network lie at relatively low elevation compared to plausible high-water levels, suggesting areas where overtopping may initiate and where more detailed site-specific assessment, including targeted monitoring, inspection or feasibility studies, would be warranted.

Nature-based solutions, such as managed dike realignment and tidal wetland restoration, offer a promising alternative. These strategies can create hydraulic buffer zones, enhance sediment deposition and support habitat and nutrient cycling functions. In the upper Bay of Fundy, several restoration projects have shown rapid ecological recovery within 5 years, including vegetation regrowth, carbon sequestration and sediment stabilization (Virgin et al., 2020; van Proosdij et al., 2023; Tackley et al., 2025b). However, implementation remains challenging in hypertidal and agriculturally productive regions like the isthmus, where land tenure, soil salinization risks and public skepticism may constrain feasibility (Sherren et al., 2016; Huguet et al., 2018). The exposure maps presented here identify broad zones where dikelands are adjacent to existing tidal wetlands or low-lying former marsh surfaces, particularly in low-elevation sections of the central isthmus, which may offer conceptual opportunities to explore nature-based approaches. For example, dike elevations are lower in the area east of the Aulac corridor (Figure 3 and Supplementary Material), making this region potentially more exposed to inundation resulting from dike overtopping. Identifying these areas does not imply suitability or preference for realignment, but rather indicates where ecological or geomorphic conditions could inform future feasibility studies.

Governance complexity is a critical barrier to adaptation. Conflicting mandates between federal and provincial governments over dike ownership and maintenance have slowed the implementation of coordinated strategies (Doucette, 2025; Grant, 2025). Funding mechanisms are inconsistent, and regulatory designations such as the Chignecto Isthmus Wilderness Area limit the types of infrastructure interventions that can occur on protected lands (Needham et al., 2020). Without a shared policy framework or joint planning authority, adaptation efforts remain fragmented and reactive.

Subsurface salinization presents an underrecognized but persistent challenge. As these impacts are not typically visible in surface-based flood mapping, they are often excluded from adaptation planning despite their long-term implications for land use and freshwater security (Tully et al., 2019). These belowground processes may not be considered in flood risk maps or policy documents, yet they may constrain recovery long after surface water has receded. Adaptation planning must navigate complex trade-offs. Hard infrastructure upgrades offer immediate protection but may be more expensive and unsustainable under long-term sea-level rise (Duc Tran et al., 2019; Vuik et al., 2019). The spatial exposure results underscore that both surface flooding and subsurface salinization risks may be concentrated in similar low-lying corridors, reinforcing the need for integrated adaptation approaches that consider both aboveground and belowground impacts.

## Socioeconomic consequences of flood exposure

Although the inundation scenarios described in the 'Evaluating flood risk through static water level monitoring' section quantify only the extent of surface flooding, these exposures have important implications for transportation, agriculture and regional economies. The following discussion highlights these socioeconomic consequences as downstream impacts rather than modeled outputs.

Effective adaptation will depend not only on technical solutions but also on governance coordination, sustained funding and inclusive engagement. Much of the at-risk land is privately owned or actively farmed, and proposed measures such as realignment or strategic retreat may face opposition without clear communication of risks and co-benefits (Needham et al., 2020; Schuerch et al., 2022). Sherren et al. (2021) highlight that the dikelands in the region are multifunctional landscapes in which both stakeholders and rights holders (e.g., rural residents, farmers, Indigenous communities) often hold differing perspectives on the value of dikelands and wetlands. These differences reflect varied cultural identities and livelihood dependencies and can lead to trade-offs between services and beneficiaries when adaptation measures are considered. Engagement with rural residents, farmers and Indigenous communities will be essential to building legitimacy and fostering support for difficult but necessary transitions (Cornejo et al., 2025). When designed inclusively, adaptation can yield benefits beyond flood mitigation, including restored ecosystem function, carbon storage and improved long-term water quality (Andrews et al., 2006; Narayan et al., 2016). Because the areas of highest modeled exposure span municipal and provincial boundaries, coordinated governance will be critical to ensuring that adaptation strategies address system-wide vulnerabilities rather than localized segments.

In recognition of these vulnerabilities, the federal and provincial governments have proposed a joint infrastructure investment to protect the corridor. As of 2025, a funding proposal of approximately $650 million (CAD) was submitted to upgrade infrastructure along the corridor (Government of New Brunswick, 2025). The project's status remains uncertain, as do pending jurisdictional agreements, cost-sharing commitments and the resolution of land use constraints. These delays mirror broader adaptation challenges facing the isthmus, reinforcing the need for a coordinated, system-wide response that addresses both short-term protection and long-term

resilience. The exposure mapping presented in this study is intended to support such system-scale planning by identifying spatially explicit areas where flood pathways, low-lying infrastructure corridors and dike elevations converge to heighten vulnerability. Any future adaptation design or prioritization would require hydrodynamic modeling, site-specific engineering assessments and detailed engagement processes that extend beyond the scope of this first-order analysis.

## Conclusions

The Chignecto Isthmus is a uniquely exposed and nationally significant coastal corridor. As the only terrestrial link between the province of Nova Scotia and mainland Canada, it underpins a critical transportation route, supports extensive agricultural activity and sustains rural communities and ecological habitats. However, much of the region lies at low elevation, protected by aging dikes that were not designed to withstand the magnitude or frequency of potential future compound high-water-level events. Current spring tides in the Cumberland Basin can already reach ~6.9 m above CGVD 2013, leaving only about 1 m between high tide levels and the mean dike crest elevation of 7.9 m and underscoring the limited safety margin. Even moderate overtopping or isolated breaches could trigger cascading disruptions to mobility, food production, groundwater quality and emergency response systems. Consistent with this, high-level static flood modeling results (Table 1) show that at water levels of 9 m AMSL, ~11 km of rail and ~ 7 km of road exist within low-elevation areas that could become inundated, magnifying the potential for interprovincial infrastructure failure and supply chain disruption. Across the modeled scenarios, static flood extents indicate that approximately 8,400–9,000 ha of cropland and up to ~660 ha of developed land are located within low-elevation areas potentially susceptible to inundation, underscoring the scale of land exposure within the isthmus corridor.

This study contributes to a growing body of work documenting the physical vulnerabilities of the isthmus and the limitations of the existing adaptation strategies. The modeled flood scenarios, based on plausible water levels, elevation data and land exposure analysis, highlight not only physical exposure but also the fragility of systems that depend on narrow bands of protection. These results represent elevation-based flood exposure estimates under plausible high-water scenarios, rather than predictions of specific hydrodynamic flood behavior, probabilities or associated damages. The findings of this study support broader calls for compound flood modeling and layered risk assessments in coastal infrastructure planning. Nature-based solutions, such as managed realignment and wetland restoration, can provide co-benefits but often require land tenure changes and risk opposition from landowners or stewarding bodies. As in other hypertidal systems, restoring ecological processes in areas of agricultural or infrastructural value remains a political and economic challenge. At the same time, fragmented governance, unclear jurisdictional mandates and regulatory constraints further limit the effectiveness of adaptation, even as climate change increases the likelihood of extreme high-water conditions. Without proactive intervention, the region faces rising exposure and diminishing recovery capacity.

As a nationally significant corridor with limited redundancy, the Chignecto Isthmus exemplifies the challenges facing low-lying infrastructure systems in a changing climate. Its combination of aging protection, concentrated exposure and governance complexity makes it a critical test case for compound flood adaptation in Canada. The consequences of inaction are not hypothetical: disruptions to transportation, food supply and freshwater access are highly plausible under current climate trajectories. A resilient future for the isthmus will require more than upgraded infrastructure. It should be framed as both a technical and a social challenge. Success will depend on a coordinated portfolio of responses that balance protection, restoration and retreat, supported by sustained monitoring, transparent governance and meaningful engagement with affected communities. The region has the opportunity to lead by example, demonstrating how layered, system-wide adaptation strategies can be designed under climate uncertainty.

Given the relative scarcity of peer-reviewed research explicitly focused on the Chignecto Isthmus, this study is intended to serve as a foundation for future work. By combining exposure modeling, land use analysis and field-based validation, it provides a starting point for researchers and decision-makers working to understand and address flood exposure in this critical coastal corridor. Future research should expand on this foundation by incorporating hydrodynamic models, assessing socioeconomic vulnerability and evaluating the long-term performance of both traditional and nature-based adaptation strategies. A coordinated research agenda, grounded in field evidence and responsive to policy needs, will be essential to supporting resilient outcomes for the region.

**Open peer review.** To view the open peer review materials for this article, please visit http://doi.org/10.1017/cft.2026.10024.

**Supplementary material.** The supplementary material for this article can be found at http://doi.org/10.1017/cft.2026.10024.

**Data availability statement.** Land use and elevation data used in this study are freely available online (see in-text citations). The processed mapping data are available from the corresponding author, H.T., upon reasonable request. The dGPS elevation data are available at https://doi.org/10.5683/SP3/1MCAQ2.

**Acknowledgements.** We thank the journal editors and two anonymous reviewers for comments that greatly improved this article.

**Author contribution.** H.T. led the conceptualization of the study, performed spatial analysis and drafted the manuscript. B.B. contributed to GIS processing, figure development and data interpretation. N.L. and B.K. reviewed and edited the manuscript for technical accuracy, clarity and contextual framing. All authors contributed to the interpretation of results and approved the final manuscript.

**Financial support.** Research funding was provided by Transformative Climate Action through an award from the Canada First Research Excellence Fund (CFREF), a Nova Scotia Graduate Scholarship to H. Tackley and grants from the Natural Sciences and Engineering Council of Canada (NSERC) through the CREATE and Alliance (OPEN-FRANC) programs.

**Competing interests.** The authors declare no conflict of interest.

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
