## [Reviewer Report]

Review of “First-Order Assessment of Flood Risk in the Chignecto Isthmus, Atlantic Canada”

Summary: Tackley et al. presents a static flood risk assessment of the Chignecto Isthmus using planar water surface modeling at 7, 8, and 9 meters above mean sea level. The study combines DEM analysis, land use classification, and field validation to evaluate exposure of infrastructure and agricultural land to flooding scenarios. While the research addresses an important and nationally significant corridor, the manuscript suffers from critical methodological gaps and inadequate presentation of key results that undermine its scientific contribution.

Comments

1. The most significant limitation of the manuscript is the complete absence of maps showing the modeled flood extents under the 7 m, 8 m, and 9 m scenarios. The reader is shown only elevation (Fig. 1), land use (Fig. 4), and a table of the results, but not the primary output on which all hazard conclusions depend. Statements regarding the vulnerability of the Aulac, Amherst, and Sackville corridors are therefore currently unsubstantiated. Because the spatial pattern of inundation is central to the results and to the policy implications that follow, the study requires a multi-panel figure set that explicitly depicts the flood extents for each modeled scenario, ideally overlaid on land use, transportation infrastructure, and dike alignments. Without these maps, the reader cannot evaluate where overtopping initiates, which basins flood first, or how inundation propagates inland, and therefore cannot assess the derived conclusions regarding exposure and adaptation needs.

2. The manuscript states that a planar flood surface modeling approach was used, but the methodological implementation is insufficiently described. It is unclear which software or spatial analysis workflow was used, how hydrologic connectivity was enforced, and, critically, how the model treated dikes. The distinction between treating dikes as impermeable topographic barriers versus ignoring them and assuming floodwater spreads once the threshold water level is exceeded will lead to markedly different flooding patterns. The treatment of these engineered features must be explicitly documented, including whether crest elevations were burned into the DEM, whether overtopping was modeled as a threshold event, or whether the DEM inherently captured crest heights. Clarifying the model configuration is essential for reproducibility and for supporting the claim that modeled inundation patterns represent plausible hydrodynamic behavior.

3. The manuscript presents the variation of dike crest elevations and notes that water levels during extreme events can come within 1.3 m of the mean crest height. However, this analysis remains disconnected from the inundation scenarios. To meaningfully relate crest geometry to flood hazard, the study should identify which dike segments lie below the 7 m, 8 m, and 9 m thresholds, and therefore constitute the likely initiation points for overtopping. A spatial visualization of crest exceedance zones would convert a system-wide average statement into a location-specific vulnerability assessment and significantly strengthen the interpretation.

4. The selection of 7 m, 8 m, and 9 m water level scenarios is described qualitatively but is not quantitatively justified. While the manuscript states that these levels are consistent with combined tide, surge, and wave setup, no explicit historical storm tide statistics or sea-level rise increments are provided. A concise contextual comparison to known historical storm events (e.g., Saxby Gale, 2015 storm tide records), along with projected sea level rise increments for 2050 and 2100, would clarify whether these scenarios represent present-day conditions, near-future hazards, or long-term climate risk thresholds. Without this contextual grounding, the scenarios appear arbitrary rather than physically motivated.

5. The reporting of inundated linear infrastructure under each scenario is an important result, but the methodological basis is unclear. It is not stated whether inundation length refers to centerline segments intersecting flooded raster cells, corridor polygons, or a simple count of partial intersections. In addition, the economic consequences of these disruptions are acknowledged but not geographically tied to the specific segments impacted. The analysis would benefit from linking the inundated corridor sections to the feasibility of detours or redundancy in the network, thereby strengthening the socio-economic argument.

6. The adaptation discussion is thoughtful, but as presented, remains somewhat general relative to the spatially explicit hazard findings that the revised figures should provide. Once flood extent maps and low-crest dike segments are illustrated, adaptation strategies could be grounded directly in the spatial distribution of exposure. This would enhance the applicability of the recommendations and clarify where opportunities for realignment or wetland restoration meaningfully exist.

7. Figure 2 (field photos) adds little analytical value and should be replaced with the missing flood extent maps

I hope you find these comments useful.

---

## [Reviewer Report]

Summary of the Manuscript:

The manuscript addresses the flood risk in the Chignecto Isthmus, a low-lying land corridor connecting Nova Scotia to mainland Canada, using a bathtub approach. The main research questions focus on evaluating the exposure of critical infrastructure and agricultural land to plausible flood scenarios and identifying barriers and opportunities for adaptation strategies under current governance conditions. The method combines a high-resolution (1 m) DEM with 30 m land-use data and extensive field surveys. The authors then simulated static uniform water levels of 7, 8, and 9 m above mean sea level, assuming no dynamic processes like wave action, storm surge, tidal timing or dike failure. Primary findings indicate that current high spring tides reach within 1.3 m of the mean dike crest elevation (7.8 m), with modest increases beyond this threshold potentially inundating thousands of hectares of cropland, hundreds of hectares of developed land and extensive linear infrastructure. These risks could jeopardize agricultural productivity (estimated $21 million CAD annual output at risk), rural livelihoods, groundwater quality via salinization, and interprovincial supply chains. The manuscript highlighting the diminishing safety margin of aging dikes, and underscoring the need for proactive, multi-jurisdictional adaptation strategies that balance hard infrastructure, nature-based solutions, and governance reforms. The manuscript addresses an understudied yet important setting of a strategic floodplain. However, the core method, static inundation modeling on a DEM, is well established. The innovation is more in application and local synthesis than in technique. The study fills a regional knowledge gap, but does not introduce fundamentally new theory or methods. Furthermore, this study highlighting the urgent need for adaptation but there is no full evaluation analysis of the effectiveness of adaptation (managed dike realignment and tidal wetland restoration) especially with water levels 7, 8, and 9 m.

In term of relevance to journal scope, the manuscript is well aligned with Cambridge Prisms: Coastal Futures. It addresses climate-driven coastal and discusses nature-based solutions and governance challenges. Also, it ties physical flood risk to socio-economic systems, infrastructure, and policy. In term of clarity and organization, the manuscript is well-organized with standard sections. The writing is clear and concise. Figures and tables support the text effectively. A few minors should be fixed (see minor comments).

Major comments:

- The authors used simplified flood modeling approach and appropriately note that their planar (bathtub) model omits dynamic processes, wave action and surge, but there is no justification on why the physics-based flood model was not used. Additionally, there is no literature reviews for the performance of bathtub model compared to hydrodynamics model in term predicted inundation extent.

- There are lacks sensitivity analysis for uncertainties. There is no uncertainty discussion on the bathtub approach and the uncertainty of DEM of 0.24m on flood impact. A sensitivity analysis on this would strengthen the method and the results.

- The choice of water levels scenarios 7-9 m AMSL is justified by past tide and surge events. It would improve clarity to specify the basis of each threshold and adding context on how these levels relate to 2100 sea-level projections and specific return periods. These revisions would help readers understand the timeframes or probabilities implied

- In results sections, the results need to be organized in better way by clearly separate the surface flood exposure from subsurface and socioeconomic impacts.

- The subsurface risks are underexplored where salinization discussion is qualitative, further analysis or model infiltration depths for robustness.

-This study highlighting the urgent need for adaptation but there is no full evaluation analysis of the effectiveness of adaptation (managed dike realignment and tidal wetland restoration) especially with water levels 7, 8, and 9 m

Figures/Tables:

- Table 1 is useful but omits uncertainty ranges (± 0.24m from DEM error), add confidence intervals.

Minor comments:

-Page 7, line 51: “Veritcal” should be “Vertical”

-Page 8, line 7: “Detetiona” should be “Detection”

-Page 15, line 8: “emergency response systems. High-level” – add comma after “systems”

---

## [Editor Report]

Dear Authors, 

two reviewers have now gone through your manuscript and provided thorough and constructive feedback, with both recommending major revisions, which I agree with. Both authors highlight important gaps in the methodological descriptions, that make the results and original contribution difficult to evaluate. Please make sure to address all comments carefully and thoroughly. Apologies for the delay in finding reviewers! We look forward to receiving your revised contribution.

---

## [Reviewer Report]

The authors have responded thoughtfully to my comments, and the changes they’ve made have noticeably improved the paper:

-The justification for the planar (bathtub) modeling approach is now much clearer, with appropriate emphasis on its suitability for a first-order, regional-scale assessment and explicit acknowledgment of the limitations relative to full hydrodynamic modeling.

-Uncertainty in the DEM and its propagation to exposure estimates has been explicitly addressed through the new sensitivity analysis and error ranges in Table 1 and the accompanying text.

-The choice of 7, 8, and 9 m water-level scenarios is now well explained in the context of present-day thresholds and compound high-water conditions.

-Results are now better organized, with clearer separation and discussion of surface inundation versus subsurface/socioeconomic impacts.

-The qualitative treatment of subsurface salinization risks is appropriately defended as fitting the scope of this screening-level study, while still highlighting its importance.

-The adaptation discussion remains appropriately scoped for a first-order assessment, with useful clarification that quantitative hydraulic evaluation is beyond the current objectives.

The manuscript is considerably strengthened by these revisions. No further revisions are required from my perspective. The paper is suitable for publication in Cambridge Prisms: Coastal Futures.

---

## [Reviewer Report]

Review of “First-Order Assessment of Flood Risk in the Chignecto Isthmus, Atlantic Canada”

I sincerely commend the authors for working hard on the manuscript. The revisions show substantial effort, and the paper is much improved. However, an important methodological framing issue remains. The static planar modeling approach, while appropriate for a first order screening analysis, represents elevation-based exposure rather than hydrodynamically simulated flood behavior. Some sections of the manuscript still describe the results in terms of flood risk in a way that could be interpreted as predictive of inundation processes. The paper would benefit from either more explicit discussion of how realistic the modeled flood pathways are, or a more consistent reframing of results as elevation-based exposure analysis.

Comments

1. The authors have done an excellent job acknowledging the limitations of the planar modeling approach throughout the manuscript. I agree that a physics-based modeling approach is not required given the stated goal of a first order assessment. However, the terminology used in key sections such as the abstract, impact statement, and conclusions should more clearly reflect that this study evaluates elevation-based exposure potential rather than flood risk in the probabilistic or hydrodynamic sense.

2. Related to this, the manuscript frequently refers to the planar model outputs as conservative. At the same time, literature cited within the paper indicates that bathtub approaches often overpredict flood extent in hydraulically unrealistic ways. These two framings are not equivalent. I recommend clarifying that the maps represent elevation-based susceptibility envelopes rather than conservative hydrodynamic flood bounds. Because the analysis does not model water depth, duration, or infrastructure performance thresholds, the results represent spatial exposure footprints rather than quantification of flood risk or damage. Making this distinction more explicit would improve scientific rigor while preserving the policy relevance of the study.

3. Section 5 provides a thorough discussion of adaptation challenges but could more directly leverage the specific spatial findings from the analysis. For example, the dike elevation analysis identifies segments most susceptible to overtopping, yet this is not explicitly connected to potential prioritization of monitoring, inspection, or feasibility studies. Similarly, the concentration of exposure in the Aulac Amherst Sackville corridor could be referenced more directly when discussing adaptation pathways. Explicitly linking these mapped patterns to potential zones of focused assessment would enhance the practical usefulness of the spatial analysis while remaining within the screening level scope of the study.

Typos/Minor edits

1. Figure 1 caption: change “diek” to “dike”.

2. p.9, ln 24: change “perdiods” to “periods”.

3. p.19, line 49: I am assuming you mean “Tackley” and not “Tackely”

4. Please use a consistent format for “sea level rise” versus “sea-level rise” throughout the manuscript.

I hope you find these comments useful.

---

## [Editor Report]

Dear Authors,

Both authors commend your extensive and thorough revisions to the manuscript in response to their comments. The manuscript is now greatly improved, especially with regard to clarifying the scope of your modeling approach and its relevance and applicability. Please address the minor revisions that R2 requests, in clarifying the language in your abstract, impact statement and conclusions to reflect the study’s focus on elevation-based flood exposure estimates, rather than risk or probabilities, and please also define what you mean by “conservative”. Please make sure to provide a point-by-point response to R2’s comments. We will be happy to accept the paper once these revisions are addressed. 

Thanks,